# Deformable image registration based on single or multi-atlas methods for automatic muscle segmentation and the generation of augmented imaging datasets

William H. Henson[1,2]*, Claudia Mazzá[1,2], Enrico Dall'Ara[2,3]

**1** Department of Mechanical Engineering, The University of Sheffield, Sheffield, United Kingdom,
**2** INSIGNEO Institute for in Silico Medicine, The University of Sheffield, Sheffield, United Kingdom,
**3** Department of Oncology and Metabolism, The University of Sheffield, Sheffield, United Kingdom

* whhenson1@sheffield.ac.uk

**Data Availability Statement:** The segmented medical images used in the registration process can be publicly downloaded from the Figshare repository: https://doi.org/10.15131/shef.data.

## Abstract

Muscle segmentation is a process relied upon to gather medical image-based muscle characterisation, useful in directly assessing muscle volume and geometry, that can be used as inputs to musculoskeletal modelling pipelines. Manual or semi-automatic techniques are typically employed to segment the muscles and quantify their properties, but they require significant manual labour and incur operator repeatability issues. In this study an automatic process is presented, aiming to segment all lower limb muscles from Magnetic Resonance (MR) imaging data simultaneously using three-dimensional (3D) deformable image registration (single inputs or multi-atlas). Twenty-three of the major lower limb skeletal muscles were segmented from five subjects, with an average Dice similarity coefficient of 0.72, and average absolute relative volume error (RVE) of 12.7% (average relative volume error of -2.2%) considering the optimal subject combinations. The multi-atlas approach showed slightly better accuracy (average DSC: 0.73; average RVE: 1.67%). Segmented MR imaging datasets of the lower limb are not widely available in the literature, limiting the potential of new, probabilistic methods such as deep learning to be used in the context of muscle segmentation. In this work, Non-linear deformable image registration is used to generate 69 manually checked, segmented, 3D, artificial datasets, allowing access for future studies to use these new methods, with a large amount of reliable reference data.

## Introduction

Muscles enable all elected movements of the human body [1]. Relationships between structural muscle characteristics such as muscle volume, geometry and length, or level of fatty infiltration and the functional capacity of individual muscles have long been established [2–4]. Specifically, muscle volume and geometry are indicative of the maximal force that a muscle is capable of outputting [3, 5, 6] and fat infiltration within muscle tissue, known as myosteosis, reduces the saturation of contractile tissue, hindering the force generating capacity of a muscle [3, 4]. Longitudinal changes in these structural characteristics are recognised as components of both

21739733. Augmented images and segmentations are also publicly available in the two Figshare repositories: https://doi.org/10.15131/shef.data. 20440164, and https://doi.org/10.15131/shef.data. 20440203, respectively. The code used to pre-process images and perform the multi-atlas segmentation can be downloaded in the Figshare repository: https://doi.org/10.15131/shef.data. 21763982, with some example inputs.

**Funding:** The study was funded by Engineering and Physical Sciences Research Council (EPSRC - https://www.ukri.org/councils/epsrc/), Frontier Multisim Grant. EP/K03877X/1, CM, EP/S032940/ 1, CM, ED. The funders had no role in study design, data collection and analysis, decision to publish, or preparation of the manuscript. There was no other external funding received for this study.

**Competing interests:** The authors have declared that no competing interests exist.

aging [7–9] and the development of musculoskeletal (MSK) and neuromusculoskeletal disorders [10–12]. Through medical imaging analysis, structural muscle characteristics are measurable *in vivo* in a process named muscle segmentation [13, 14].

Both Computed Tomography (CT) and Magnetic Resonance (MR) imaging, have been used to non-invasively gather quantitative structural muscle characteristics such as volume [15, 16] or geometric shape [4, 16, 17]. The structural characteristics of the skeletal muscles within the lower limb are of particular interest, due to their capacity to enable locomotion [1, 15, 18]. As the lower limbs are such a large area of the body, many studies prefer MR imaging over CT, to limit ionising radiation exposure of subjects enrolled in studies or of patients in future potential clinical applications [15, 18]. The current approach used within the literature to gather these structural muscle characteristics from MR images is manual segmentation, during which the operator defines in each slice of the MR imaging sequence (or in a subgroup of them) the contour of each muscle [15, 18, 19]. There are two main limitations of manual segmentation: the required operator input time and operator dependency issues of the outputs [6, 15, 20]. The high processing time is incurred as there are around 35 muscles within an individual lower limb [15, 21] that must be manually segmented from in the order of hundreds of images. Recent advancements in computer vision (interpolation between segmented slices) and hardware (trackpads) have decreased operator interaction time down significantly, to approximately 10 hours [15, 19, 20]. Not only is this interaction time still excessive, but operators must undergo training to achieve repeatable segmentation results from an intra-operator standpoint (± 10% volume is typically acceptable) [15]. Regardless of training, as suggested, there are significant inter-operator dependency issues noted within the literature, which have been shown to misinterpret muscle volume by up to 50% (for example, the peroneus brevis and longus [15]), depending on the muscle of interest and study cohort [15, 20, 22]. These limitations of the gold standard approach prevent the utilisation of muscle segmentation as a technique to inform large-scale quantitative investigations into muscle characteristics.

Many different automatic segmentation methods have been investigated within the literature in recent years to replace the manual approach [13, 17, 20]. Image registration is one that has been explored within the literature to perform muscle segmentation [23–25]. Simplistic applications, such as two-dimensional (2D) deformable image registration between subsequent MR imaging slices within subjects has been used to propagate segmentations of individual slices into partial sections of 3D muscle geometry using only a few manually segmented slices, with encouraging results (DSC ≈ 0.91) [23]. 3D image registration has been used within longitudinal studies to populate MR images with partial segmentations of a small number of muscles to good effect, such as within the studies presented by Le Troter et al. [24] and Fontana et al. [25]. Though this longitudinal approach provides insight into the change in muscle characteristics over time, multiple MR image sequences are required from individual subjects at two different timepoints and one dataset must be manually segmented. These studies all operated different registration algorithms, such as antsRegistration [23], NiftiReg [24], or ITK-snap [25]. Another elastic registration algorithm used for image segmentation or registration of hard and soft tissues with a high degree of accuracy is the Sheffield Image Registration Toolkit (ShIRT) [26–28]. Although ShIRT has features that enable the tracking of large deformations and potential changes in grey-levels between the fixed and moved images, ideal for evaluating changes in size and shapes of soft tissue, it has not yet been tested in the context of muscle segmentation. https://doi.org/10.15131/shef.data.20440203.

Moreover, multi-atlas registration-based methods have been used in the context of muscle segmentation [29, 30]. Multi-atlas registration-based methods typically follow a similar structure, wherein multiple medical imaging datasets are registered to a target image, and a probability map is built [31, 32]. The probability map defines the probability that each pixel (or

voxel) in the target image belongs to a certain class, in our case the classes would be the different muscles. If the probability map agrees to a certain threshold, a pixel is defined to belong to a certain class. The disputed pixels, that fall below the threshold are labelled corresponding to the registered image that is the most similar to the target image. Highly successful in this regard, was the full body muscle segmentation performed by Karlsson et al. [29], where muscles groups within the arms, torso, thighs, and calves were segmented. A multi-atlas operation was used to automatically segment muscle groups with a moderately high degree of accuracy, capturing the volume of these muscle groups automatically with error lower than 15%.

Single input and multi-atlas registration-based methods have been used to good effect in the context of medical image segmentation [31, 32], and more specifically muscle segmentation [25, 30], but they have not yet been fully explored. Explicitly, inter-subject registration aiming to segment all individual muscles within a new subject, referencing a previously segmented subject has not yet been explored to the best of the author's knowledge. Additionally, combining the outputs of distinct registrations in a multi-atlas fashion is yet to be explored for the segmentation of individual muscles simultaneously. These methods would be of great interest as it would allow already existing databases of segmented images to be propagated to newly acquired imaging datasets.

Probabilistic machine learning methods such as deep learning have been used to automatically segment the 3D geometry of individual muscles from MR images taken from several different cohorts [19, 20, 22]. These methods employ Convolutional Neural Networks (CNNs) which learn patterns that identify important features from training data in order to apply these learned patterns to segment new, unseen data [19, 20, 22]. Notably, the methods recently proposed by Ni et al. [22], where all lower limb muscles within a cohort (n = 64) of young healthy athletes were segmented with DSC comparable to that of the inter-operator dependence (DSC ≈ 0.9), and those proposed by Zhu et al. [20] where all muscles within the shank were segmented from a cohort (n = 20) of children with cerebral palsy (DSC ≈ 0.88). Though the segmentation accuracy found within these studies is remarkable, these methods are not widely accessible due to the main limitation of current deep learning methods: the requirement of large training databases (minimum ~20 segmented 3D images, the greater this number the more robust the method) [33]. Unfortunately, generating these segmented imaging datasets might not be well suited to MR imaging, given the associated high costs and manual processing time. Additionally, when used in medical image segmentation, deep learning generally has the major limitation of a significantly reduced performance when assessing imaging data taken from widely varying cohorts [33]. Data augmentation is a technique widely used in association with CNNs for the purpose of supplying greater amounts of training data and helping to generalise their application to image classification and segmentation tasks [34, 35]. Within this context, image registration has been previously used to generate augmented images to facilitate the analysis of brain tumours [36] and skeletal deformities [37]. This suggests that, while not attempted before, similar approaches might be adopted for muscle segmentation.

The aim of this study is hence twofold. The first is to evaluate the accuracy of a novel method for automatic segmentation based on single input or multi-atlas of complete 3D geometry of most individual skeletal muscles in the lower limb from MR imaging data simultaneously using 3D deformable image registration. Secondly, the effectiveness of deformable image registration in the generation of augmented datasets is explored and the benefits of this highlighted.

## Methods

### Subjects & imaging acquisition method

Retrospectively available lower limb T1-weighted MR images from 11 post-menopausal women (mean (standard deviation): 69 (7) years old, 66.9 (7.7) kg, 159 (3) cm) were used for this study

[15]. Images were collected using a Magnetom Avanto 1.5T scanner (Siemens, Erlangen Germany), with an echo time of 2.59 ms, repetition time of 7.64 ms, flip angle of 10 degrees. The study was approved by the East of England–Cambridgeshire and Hertfordshire Research Ethics Committee and the Health Research Authority (16/EE/0049). The MR images were acquired in four sequences, capturing the hip, thigh, knee, and shank. To reduce scanning time while still providing detailed geometries of the joints for use within the original study, the joints were acquired with a higher resolution (pixel size 1.05 mm$^2$, slice spacing 3.00 mm) than the long bone sections (pixel size 1.15 mm$^2$, slice spacing 5.00 mm). The sequences were stacked in MATLAB forming one continuous 3D image from hip to ankle, firstly by homogenising the resolution of each of the imaging sequences taken from the different sections to be 1.00x1.00x1.00 mm$^3$ through tri-linear interpolation (interp3, MATLAB 2006a). The fields of view of the images across the four sequences were equated by wrapping the images in blank data (greyscale value of 0), referencing the spatial metadata of the images to retain the relative subject position across the imaging sequences for each subject. The homogenised sequences were concatenated in the longitudinal direction, removing half of any overlapping volume from each section where the fields of view overlapped. The removal of half of the overlapping volume from each of the sequences also removed the images affected by MR imaging bias. A bias field correction algorithm was tested but did not alter the images [38]. Lastly, the images were cut in half in the frontal axis, isolating only the right limb and the field of view was reduced such that the images contained only the anatomical data. A sub-cohort of 5 of the 11 subjects was selected for automatic segmentation. The five subjects were chosen with the aim of creating a sub-cohort with a wide anatomical diversity, choosing the shortest and tallest (154.0 cm, 164.2 cm), the subjects with the lowest and highest Body Mass Index (BMI, kg/m$^2$) (21.2, 32.1), and the youngest and oldest participants (59, 83). Each subject was used as both a target and a reference for the image registration algorithm, creating 20 subject pairings for the sub-cohort (inter-subject analysis). For comparison, the 5 subjects were registered with a procedure similar to the inter-subject analysis, using the opposing limb (left vs right) as the reference dataset for the registration (intra-subject analysis).

## Reference segmentations

Each of the five subjects involved in this study were segmented manually, as presented by Montefiori et al. [15]. Within this database, the muscles for which the coefficient of variation of the manual segmentations when repeated by the same operator on three separate runs was greater than 10% were removed from the study, reducing the number of muscles considered in this study from 35 to 23. Table 1 presents the range of volumes of the 23 muscles considered within this study within the cohort of 5 subjects. Their manual muscle segmentations were used as the templates to populate imaging data of new subjects with automatically generated muscle segmentations through image registration and to validate them. To aid the interpretation of results, the variability of each of the muscles was calculated as the ratio of the range and mean volumes within the cohort.

The range of volumes of the muscles included within the study for the 5 subjects considered. The muscles are separated into three sections of the body (hip, thigh, and shank). The muscles considered are those that were segmented with an acceptable level of repeatability [15]. The variability of muscle volumes within our cohort (calculated as a ratio of the range to the mean) is highlighted. Full description of muscle volumes within each subject expanded upon in S1 File.

## Image pre-processing

Initial registration experiments of imaging sequences from two different subjects showed that the difference in the thickness of the fat surrounding the muscle tissue skewed the registration

**Table 1. Variability of muscle volumes.**

| Body segment | Muscle | Volumes | | Variability |
|---|---|---|---|---|
| | | Minimum | Maximum | Range/mean |
| | | (cm$^3$) | (cm$^3$) | |
| Hip | Adductor brevis | 54.2 | 67.1 | 21.3% |
| | Adductor longus | 59.7 | 91.7 | 44.6% |
| | Adductor magnus | 282 | 457 | 49.2% |
| | Gluteus maximus | 406 | 654 | 45.4% |
| | Iliacus | 81.8 | 127 | 41.3% |
| | Tensor fasciae latae | 17.4 | 57.9 | 95.8% |
| Thigh | Biceps femoris caput brevis | 31.5 | 80.7 | 78.1% |
| | Biceps femoris caput longum | 95.3 | 128 | 28.6% |
| | Gracilis | 37.6 | 51.2 | 30.5% |
| | Rectus femoris | 94 | 125 | 27.1% |
| | Sartorius | 62.7 | 105 | 53.3% |
| | Semimembranosus | 98.9 | 154 | 45.6% |
| | Semitendinosus | 88.5 | 101 | 13.2% |
| | Vastus intermedius | 214 | 313 | 38.4% |
| | Vastus lateralis | 303 | 351 | 14.7% |
| | Vastus medialis | 167 | 277 | 51.3% |
| Shank | Gastrocnemius lateralis | 78.2 | 87 | 10.6% |
| | Gastrocnemius medialis | 123 | 176 | 34.6% |
| | Peroneus brevis | 33.7 | 41.6 | 20.2% |
| | Peroneus longus | 25.7 | 59 | 88.5% |
| | Soleus | 304 | 406 | 30.6% |
| | Tibialis anterior | 74.4 | 94.2 | 23.6% |
| | Tibialis posterior | 56.3 | 90.6 | 45.8% |

and resulted in a poor registration quality (see S5 File). In order to homogenise this feature, the MR images of each subject were pre-processed to homogenise the distribution of fat tissue within the scans, focussing the registration on the muscle tissue. For each 2D slice of imaging data (example slice shown in Fig 1A) within each subject, firstly, the air-skin boundary was

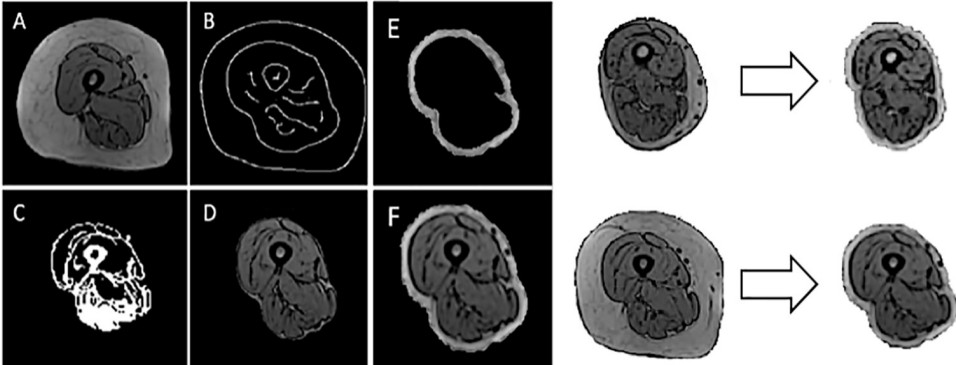

**Fig 1. Homogenisation of the fat tissue surrounding the muscles.** The process of masking the fat tissue surrounding the muscles from the raw MR images (left) and wrapping in a homogenous layer of fat for two images taken from different subjects (right). The subject along the top row (right) had a fat layer less that 5 mm thick and was wrapped with artificial fat, where the subject along the bottom row had a depth that was sufficient.

located using a Canny edge detector [39]. The area within the skin boundary was filtered (Fig 1C), in response to a threshold established from the greyscale frequency intensity plots of the images, creating a mask that contained only the muscle tissue (Fig 1D). A layer of fat was wrapped around the muscle tissue (Fig 1E and 1F) to emphasise the outer boundary of the muscle tissue. The depth of this layer of fat was made equal to the optimal nodal spacing (NS, a parameter of the registration [26], set to 5 mm, details in 2.4) as the registration operates optimally in the circumstance that the object being registered is of similar size to the NS [26]. There were two possible scenarios for the fat wrapping process: 1) the layer of fat within the image was greater than 5 mm, and 2) the layer of fat was less than 5 mm. In the first scenario, the subject's fat tissue was wrapped around the muscle tissue at a depth of 5 mm. In the second scenario, artificial fat was wrapped around the body which was built in response to the greyscale frequency intensity peak that represents the fat. The pixels within 5 mm of the muscle tissue that lay outside the body were randomly assigned values using a uniform distribution with minimum and maximum equal to the mean ± standard deviation of the frequency intensity peak representing the fat. Through this operation, the muscle tissue remained unchanged, but the fat tissue surrounding the muscle was reduced, meaning that all muscle characteristics (muscle volume, shape, and fat infiltration) are all conserved.

### Registration & automatic segmentation

Following pre-processing, subject imaging datasets were registered using an in-house deformable image registration algorithm, ShIRT [26]. ShIRT performs deformable (non-linear) image registration, allowing high degrees of anatomical variability between inputted images to be addressed [26–28], and has the potential to automatically segment blocks of muscles or individual muscles given a fully segmented reference subject, but this is yet to be explored. In the registration process, displacement functions were computed that map each pixel in a reference image to a corresponding pixel in the target image, with no initialisation. The registration is performed by iteratively reducing a cost function, which represents the sum of squared differences between the intensity values within the images. ShIRT solves displacement equations at nodes of an isotropic hexahedral grid overlapped to the fixed and moved images, with distance between the nodes equal to NS. The optimal NS for this registration task was found through a sensitivity analysis (see S2 File). Throughout the registration process the optimal nodal displacements are smoothed in response to a smoothing coefficient, optimised in each registration to solve the registration problem [26] (this was verified to be indeed optimal for this application, see sensitivity analysis in S2 File). The 3D displacement field is calculated using tri-linear interpolated displacements between the nodes of the grid. The registered image was generated after applying the transformation to the reference image and using tri-linear interpolation. Similarly, the automatic segmentation of the muscles within the target subject was calculated applying the transformation to the manual segmentations of the reference subject (Fig 2).

To gauge the accuracy of the resulting segmentations, the registration and segmentation pipeline was used to segment the right limbs of the 5 subjects using the contralateral limb as the reference input. The muscles within opposing limbs have been proven to be anatomically similar but distinct in both volume and geometry [15]. For these reasons, using the opposing limb in the segmentation pipeline should provide the best possible reference for the segmentation of the muscles within each of the 5 subjects.

### Multi-atlas segmentation

Following the registration between the images of the 5 subjects, the resulting segmentations and registered images were used to define a multi-atlas segmentation for each muscle within

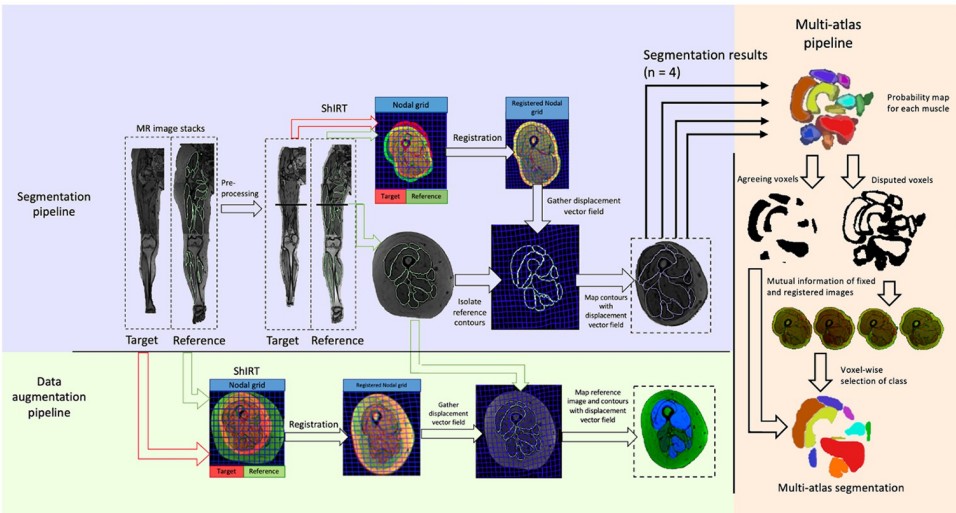

**Fig 2. Registration, multi-atlas, and image augmentation pipelines.** The image registration process, shown for one 2D slice of imaging data (location within imaging sequences highlighted with a black line). Segmentation pipeline: the target and reference subject were pre-processed, homogenizing the fat layer, and registered in ShIRT. The map found through registration was applied to the manual segmentation contours of the reference subject (shown in green), resulting in an automatic segmentation of the target subject (shown in blue). Data augmentation pipeline: The combined MR imaging sequences are registered in ShIRT. The map outputted from the registration was used to deform the reference subject's 3D imaging data and reference manual segmentations, resulting in a fully segmented, augmented 3D image. The augmented images are shown with each muscle taking a different greyscale value (visualised in blue image channel). The multi-atlas pipeline used is also shown on the right-hand side. The outputs of the segmentation pipeline are combined forming a probability map for each muscle. The disputed and agreed voxels are separated, and the disputed voxels are labelled according to the best performing registration, found through calculating the mutual information of registered and fixed images. Collating these results provides a multi-atlas segmentation for each muscle.

each subject. A probability map was defined for each voxel in each of the target images, representing the probability that a given voxel belongs to a certain muscle. The probability map was created by adding the segmentations from each of the four references together. The overlapping portion of all four segmentations for a given muscle was assigned a probability value of 1 and were included in the multi-atlas segmentation output. The disputed voxels (those that did not have a probability of 1), for any given muscle were assigned to a given class, based on the localized mutual information between each of the registered images and the target image, in a method similar to that of Gholipour et al. [40]. The localized mutual information was calculated voxel-wise between registered and target images as the sum of squared differences (the similarity measure used within the registration algorithm) in a $25^3$ voxel volume surrounding each pixel [31]. The registered image presenting the maximal agreement with the target image (that with the lowest sum of squared differences) at each of the disputed voxels was found, and the segmentation of that voxel resulting from its registration was selected.

## Segmentation validation

The registered reference and target images were overlapped to assess the quality of the registration. The two images were visualised simultaneously, with the registered and target images shown in green and red, respectively. Well registered images appear yellow with very few green or red flecks. Fig 3 presents three example registration results, where the quality of registration increases from left to right.

**Fig 3. Sample registration outputs for qualitative interpretation.** Registration results of images taken from the shank. The registration quality is visualised within these plots with poor, moderate, and flawless registrations shown in a, b, and c, respectively. Yellow colour represents well registered regions.

Three complementary quantitative metrics were used to test the accuracy of the automatic segmentation protocol. The relative volume error (RVE) was calculated following Eq 1 for each muscle in each subject. Additionally, the total volume error (TVE) between the reference and automatically segmented muscles was calculated as the error between the sum of all muscle volumes, shown in Eq 1.

$$RVE_{i,j} = 100 \times \frac{V_{A_{i,j}} - V_{M_{i,j}}}{V_{M_{i,j}}}, TVE_j = 100 \times \frac{\sum_{i=1}^{N} |V_{A_{i,j}} - V_{M_{i,j}}|}{\sum_{i=1}^{N} V_{M_{i,j}}} \qquad (1)$$

Where $V_{Ai,j}$ and $V_{Mi,j}$ are the volumes of the automatic muscle segmentation and ground truth (manual) segmentations, respectively.

The Dice similarity coefficient (DSC) [41] was used to assess the accuracy of segmentation considering both volume and geometry, through comparison with the ground truth segmentation. The DSC varies between 0 and 1, with a value of 1 signifying that the proposed segmentation and ground truth are identical. The DSC was calculated (Eq 2) for each muscle (i) in each subject (j), where $A_{i,j}$ and $M_{i,j}$ represent the automatic muscle segmentation and the ground truth segmentation, respectively.

$$DSC_{i,j} = \frac{2\left(A_{i,j} \cap M_{i,j}\right)}{|A_{i,j}| \cup |M_{i,j}|} \qquad (2)$$

Finally, the Hausdorff distance (HD) [42] between the automatic and reference muscle segmentations was calculated for each muscle in each subject, following Eq 3, where $a_{i,j}$ is an element of $A_{i,j}$, $m_{i,j}$, is an element of $M_{i,j}$, and $d$ is the magnitude of the minimum distance between $a_{i,j}$ or $m_{i,j}$ and the nearest neighbouring point within $M_{i,j}$ or $A_{i,j}$, respectively. For each subject the HD was calculated as the maximum among the minimum distances between the automatic and ground truth segmentations.

$$HD\left(A_{i,j}, M_{i,j}\right) = max\left\{\left(d\left(A_{i,j}, m_{i,j}\right)\right), \left(d\left(a_{i,j}, M_{i,j}\right)\right)\right\} \qquad (3)$$

## Generation of augmented data

The deformable image registration algorithm was used to generate segmented augmented MR imaging data, available for download within S3 File. The stacked MR imaging data from the right limb of the 11 participants were registered to each of the other subjects in the cohort, giving 110 combinations. No pre-processing was applied. The displacement vector field outputted

from ShIRT (Fig 2) was used to deform both the MR imaging sequence and the manual muscle segmentations of the reference subject. The output of each of these processes was a fully segmented 3D image that was dissimilar to both the reference subject and the target subject (Fig 2). A four-point criterion was used for checking both the images and the segmentations to ensure anatomical credibility of the augmented dataset: a) the boundaries of the long bones and the skin must be reasonably smooth and continuous; b) the positioning and orientation of the joints must be anatomically viable, with the bones fitting together realistically; c) the muscle segmentations should reflect the muscle structure; and d) the location of each of the muscles relative to one another must be realistic (e.g. the vastus lateralis must be lateral with respect to the vastus medialis). If any one of these criteria were not met, the augmented dataset was discarded. Out of the retained datasets, 15 chosen at random were retested by a different operator to confirm the specificity of the inclusion criteria. Finally, the available muscle volumes were compared from within the augmented and original databases. The mean volume within each database was computed for each of the 23 muscles considered. The difference between the volume of each muscle within the database and the average was then calculated, and this value was normalised against the mean volume. The resulting values were percentages representing the distribution of available muscle volumes within each database, which after normalisation, could be compared.

## Results

### Segmentation results

A visualisation of an example registration and of the results of one segmentation are highlighted in Fig 4 for images taken from the hip, thigh, and shank, respectively. While the deformable image registration has accurately identified the muscle tissue in the target subject in most cases (yellow), some regions were not correctly registered (red or green). The segmentation results reflect this, where the registration appears successful overall, and the automatic segmentations are geometrically very similar to the reference segmentations. There are areas within the automatic segmentations that do not reflect the reference segmentations, such as the gluteus maximus in the hip section, and the tibialis muscles within the shank section. The automatic segmentations within the thigh section mostly agree with the reference segmentations.

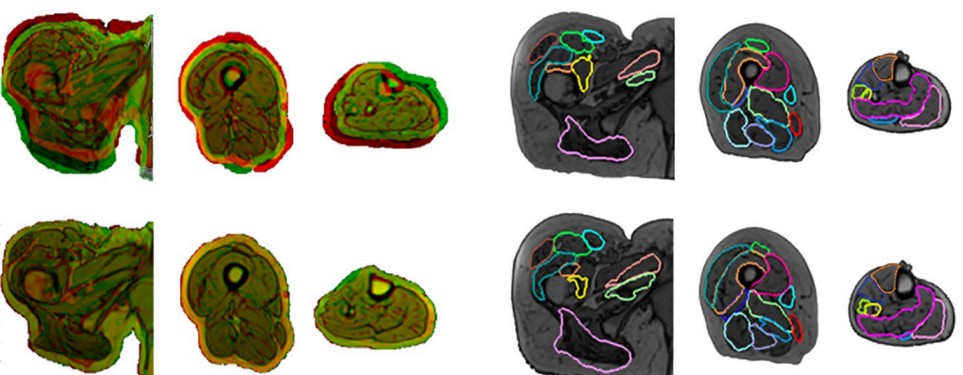

**Fig 4. Qualitative interpretation of segmentation results.** Registration and segmentation results from the combination of subjects resulting in the median average DSC (subject 4 and 2 as the target and reference, respectively). The registration inputs (top row) and outputs (bottom row) for these combinations of subjects are shown on the group of images on the left. The segmentation results are shown in the right three image groups, where the reference and automatic segmentations for the target subject are shown in blue and red respectively. The muscles that are not highlighted within the images, were found not to be segmented with an appropriate level of repeatability.

**Volume error.** The TVE for the entire muscle body was 8.2 ± 5.1% (mean ± standard deviation) across all subject combinations (Fig 5). The mean RVE for the individual muscles was found to be below 12.8% for all combinations and all upper quartiles were below 40% error. The best performing combination was subject 5 with 1 as the target and reference respectively, with among the smallest mean (-2.2%) and with the lowest quartiles (lower and upper quartiles of -10.5% and 6.4%, respectively). The relative volume error was consistent across all muscles, with no correlation found between muscle volume and relative volume error ($R^2$ = 0.092, p-value = 0.159); the muscles with the highest variability within this cohort (tensor fascia latae, rectus femoris, and peroneus longus) made up the outliers within the distributions of RVE, as the registration algorithm was unable to overcome the large differences in volume. The mean RVE from the left to right analysis was 0.35% (Fig 5), similar to but outperforming the best inter-subject results (3.2%). The multi-atlas analysis provided a lower inter-quartile range in terms of RVE and resulted in the mean RVE across the 5 subjects falling within the acceptable range of error (range of means = [-2.4, 9.0] %), which cannot be said for the single atlas registration results (range of means = [-17.8, 14.2] %).

**Dice similarity coefficient.** When looking at the segmentations of the five subjects obtained using the other four as reference subjects, very variable results were observed. The greatest average DSCs were those resulting from the segmentation of subjects 1, 2, and 4, using subject 2, 1 and 1 as the reference subject, respectively. The mean DSCs found for these combinations of subjects were greater than 0.70, lower quartiles greater than 0.67, and with a wide spread of results ($0.35 < DSC < 0.88$). Subjects 3 and 5 were segmented with a consistently lower DSC, with the average DSC considering all reference subjects found to be 0.61 and 0.60 respectively (0.69, 0.69 and 0.67 for subject 1, 2, and 4, respectively). Additionally, one of these subjects was always the worst performing reference subject considering DSC when used to segment all target subjects, with the lowest average DSC. There was a weak correlation found between muscle volume and the DSC of the automatic segmentations ($R^2$ = 0.332, p-value = 0.003), suggesting that the larger muscles were slightly better segmented in terms of DSC. The average DSC found within the intra-subject analysis was 0.80 (Fig 5). Finally, the multi-atlas segmentation results presented a more consistent DSC for each of the subjects used, with a lesser range and a higher lower quartile value but with comparable average DSC.

**Hausdorff distance.** Overall, the average HD was typically between 15 mm and 30 mm, with the upper quartile being below 40 mm, other than the segmentations of subject 3 and 5 using subject 2 and 3 as references, respectively (Fig 5). The spread of results was large, with Interquartile ranges (IQR) being between 7 mm and 21 mm. There was no correlation found between the HD and the size of the muscle for which the HD was calculated ($R^2$ = 0.097, p-value = 0.089), the error was consistent across muscles of all sizes. The average HD found within the intra-subject analysis was 17.7 mm, much lower than in the other analyses. The HD distances in the multi-atlas analysis were comparable to the inter-subject results. Subject 1 and 2 were segmented with a lower HD in the multi-atlas approach compared to the optimal inter-subject combinations, and subject 3, 4, and 5, slightly worse than the respective optimal inter-subject combinations.

## Augmented data

After initial checking by the author, 69 of the 110 generated augmented datasets passed the inclusion criteria. 15 datasets were rechecked by an expert in muscle segmentation and all 15 passed, giving 100% specificity. Fig 6 showcases some examples of the augmented images collected. Visually, the augmented images are well segmented, and are dissimilar to the reference subjects, particularly in the second row of images, where the relative fat depth of the moving

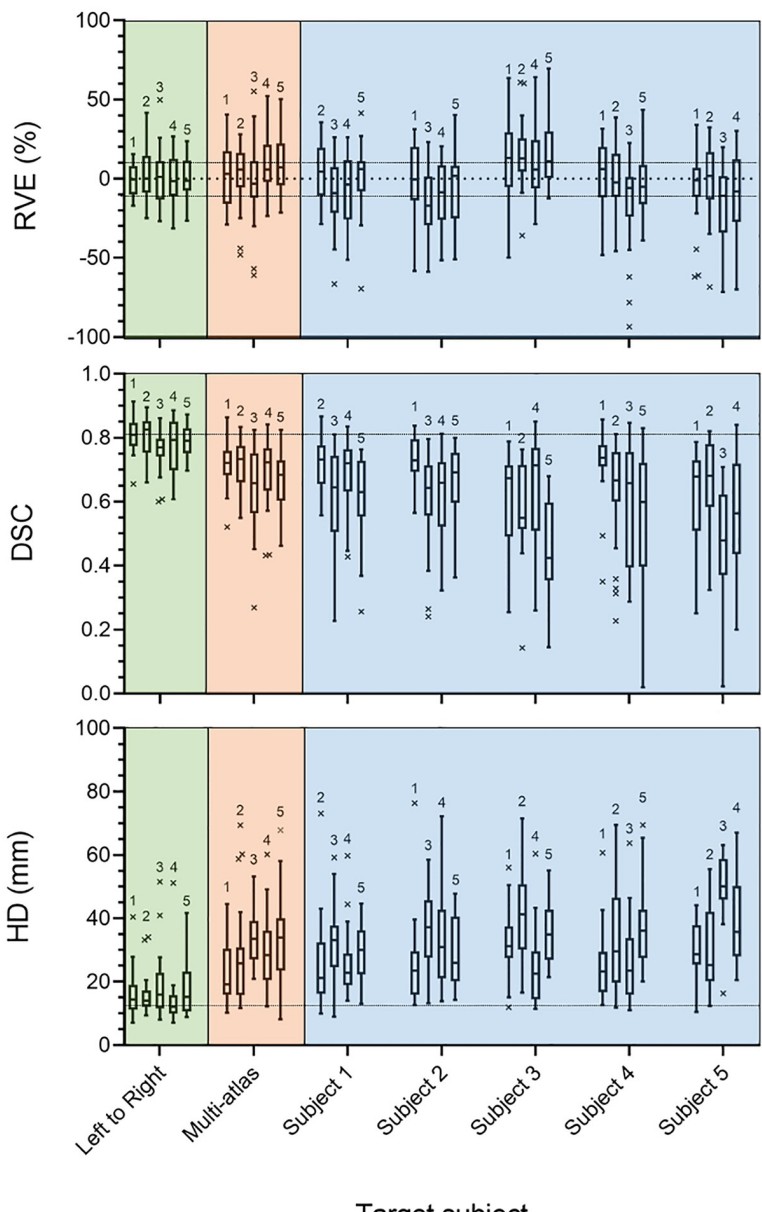

**Fig 5. Numerical results for the intra-subject, multi-atlas, and single atlas analyses.** Relative volume error (%) (top), Dice similarity coefficient (centre), and Hausdorff distances (mm) (bottom) found for each muscle in each subject, for the left to right analysis (green), multi-atlas (orange), and inter-subject approaches (blue). In both the left to right and multi-atlas analysis, the numbers above the boxplots denote the subject segmented. The numbers above each of the boxplots in the inter-subject approaches denote the reference subject used within the registration. The dashed line in RVE plot shows the acceptable level of RVE resulting from inter-operator dependence, prescribed by Montefiori et al. 2020 [15]. The grey dashed lines in the DSC and HD plots represent the mean values from the intra-subject analysis for comparison. The box and whisker plots show the mean, interquartile ranges, and ranges across the 23 muscles considered.

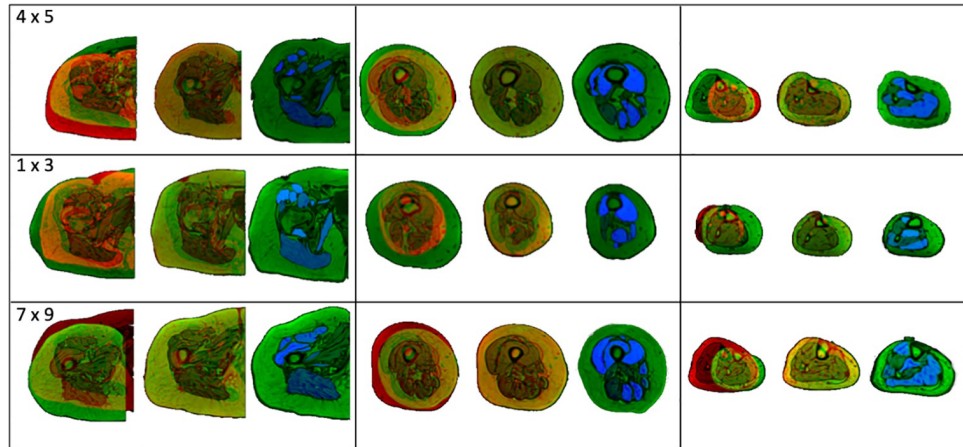

**Fig 6. Exemplar augmented images.** Inputs, outputs and resulting augmented subjects. Each row of images presents results within the hip (left), thigh (centre), and shank (right) for 3 subject combinations chosen at random (target x reference: 4 x 5 (top), 1 x 3 (middle), 7 x 9 (bottom)). Within each cell there are the inputted images into the registration (left), registered images with corresponding target image (centre) and resulting segmented, augmented images (right). The muscle labels are visible within the augmented images as the blue areas. Each muscle is assigned a distinct greyscale value and the labels are assigned alphabetically.

subject (green) is retained, but the cross-sectional area of the thigh is equated to the fixed subject (red). The misalignment of the muscle tissue within the registered images, visible as concentrations of either red or green colouration, establish a difference in the muscle geometry within the registered and original data. The augmented subjects generated for 1 target subject (subject 1) are presented within S3 File, for visual comparison.

The anatomical variability of the muscles within the augmented database is compared to the original 11 subject database (Fig 7). The volumes of each of the muscles within the original and augmented databases were normalised against the corresponding average muscle volume for each muscle within the respective databases. The percentage greater or smaller than the average volume was then calculated for each muscle, representing the variability of the muscle volumes within each database. The distributions of these percentages are presented (Fig 7). The muscle volumes available within the augmented database were found to have a greater range of volumes, often 1.5 to 2 times greater than in the original database. The range of volumes for each muscle considered within the original and augmented databases are presented in S4 File.

## Discussion

This paper aimed at proposing a fully automatic tool to segment 23 major lower limb muscles simultaneously from MR imaging data using morphological image processing, deformable image registration and a multi-atlas approach. Furthermore, the registration tool was used to generate a unique dataset including 69 fully segmented, augmented 3D images. To the best of the authors' knowledge, this study represents the first attempt to segment complete 3D muscle geometry of many individual muscles simultaneously using deformable image registration while using different subjects as the reference. Moreover, a multi-atlas approach was used for the segmentation of many individual muscles simultaneously, which is yet to be investigated in this way. A well performing automatic segmentation tool would be desirable as muscle segmentation of a new subject could be performed automatically, without the need for manual processing.

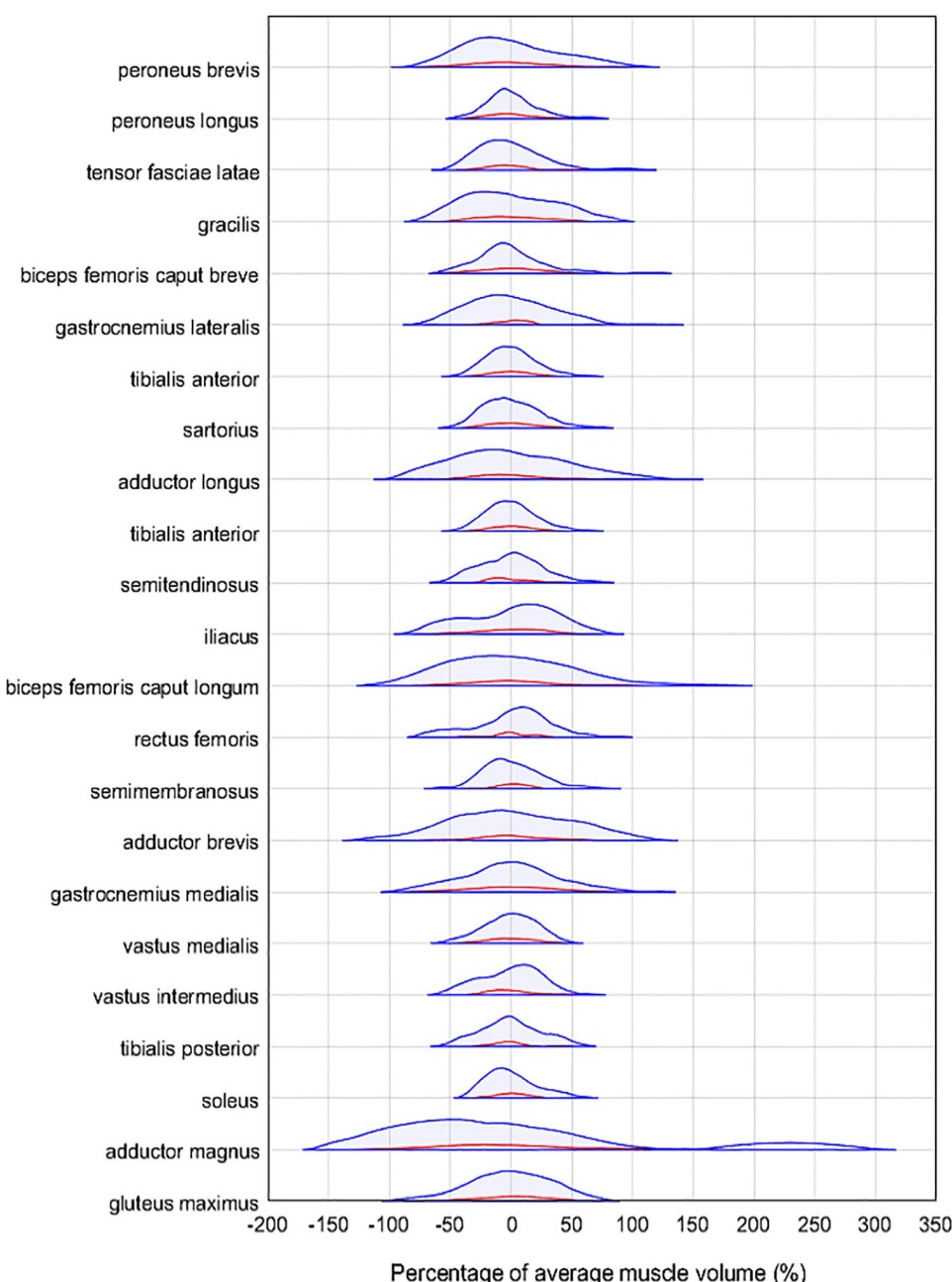

**Fig 7. Enhancement of muscle volume variability through image augmentation.** The anatomical variability of muscle volumes for each muscle, ordered from smallest to largest within the original and augmented databases shown in red and blue respectively. The height of the distributions was not normalised, and the violin plot contains 95% of the data, with 2.5% of data cut off from each side, removing outliers.

All 23 muscles were segmented from five subjects with moderate success, considering three error metrics, the RVE, DSC and HD. The registration quality was high considering the combination of subjects that resulted in the median average DSC (Fig 4) which suggests that in most cases, the registration performed as intended. This was confirmed by the total volume error metric, lower than 10% on average. However, all three-error metrics reflected a lower accuracy for the segmentation of individual muscles. Within both the inter-subject and multi

atlas analyses, the individual muscle RVE was typically larger than that of an acceptable level of inter-operator dependence (±10%) [15], with one or both of the lower and upper quartiles often exceeding ±10% in most subject combinations. The mean absolute RVE within the optimal subject combinations was 12.7%, meaning that on average, there was an over or underestimation of the muscle volume greater than that incurred by the effects of operator variability. This indicates that the method would be best suited when only interested in the volume of the overall muscle body. Capturing the total muscle volume has proven useful in studies such as Handsfield et al. [18], where regression equations were presented, to estimate individual muscle volume from total muscle volume and other anthropometric data such as height and BMI. Additionally, this result could be useful for an estimation of the level of fatty infiltration into the muscle body [8]. The DSC results, on the other end, indicate that if the purpose of the segmentation was that of extracting internal muscle characteristics, such as the level of fat infiltration [10], then alternative approaches should be pursued regardless of the inclusion of a multi-atlas postprocessing step. Possible improvements of the method could come from a more targeted selection of the reference subject, which as shown by the reported results (Fig 5) can increase the accuracy of the approach both in terms of individual muscle volume and DSC. Though, it is extremely likely that a better reference than a subject's contralateral limb would be seldom found as an input of the registration algorithm. Therefore, the use of deformable image registration of images acquired with these acquisition parameters for the purpose of individual muscle segmentation could be segmented at best with an accuracy comparable to that of the left to right analysis (~0.8 DSC).

The geometry of each of the 23 muscles was captured moderately well for the optimal combination of subjects in the inter-subject analyses (those with greatest lower quartile), with mean DSC of 0.74 and IQR range of $0.71 < DSC < 0.77$ and the multi-atlas approach presented very similar results. However, these quantitative measures of accuracy are significantly lower than the inter-operator dependence of the manual process, which, within the literature [2, 15, 20, 22] is consistently found to be DSCs of around 0.90 for the muscles considered in this study. While the pair of subjects leading to the best results in terms of DSC were the most similar in terms of height and BMI, these anthropometric characteristics were very different in the pair having the second-best DSC (mean = 0.74, IQR of $0.69 < DSC < 0.79$). This suggests that the newly proposed masking process achieved the goal of homogenising the subject imaging data and could be adapted for the removal of unwanted artefacts from within medical or indeed any other images. The worst performing combination of subjects (those with the lowest upper quartile), with mean 0.45 DSC across the 23 muscle segmentations, were those with the greatest difference in age (16 years) but similar height, weight, and BMI. One could suggest that the muscle quality between these two subjects could be the greatest, in response to age-related degradation of the muscle tissue [43]. The muscle quality greatly affects the appearance of the muscles within medical images and would certainly affect the quality of the registration [43].

Particularly successful approaches within the literature that used 3D deformable image registration to perform muscle segmentation were those based on longitudinal data, such as Le Troter et al. [24] and Fontana et al. [25], who attained average DSCs of 0.90 and 0.85, respectively. Similar to the latter were the DSC values here found when registering the left to right limb in the same subject, which could be used to propagate segmentations of one limb to the contralateral limb, halving the time to segment the lower limbs. Notably, the approaches found within the literature still require the manual segmentation of each subject at the baseline. Moreover, these studies segmented fewer muscles than the 23 presented in this study, presenting a reason for the reduced accuracy in this study. Last but not least, the images collected within this study were not optimised for muscle segmentation, as they were the widely used

T1-weighted MR images with lower resolution images acquired along the long bone sections (see section 2.1). Both characteristics of the images used may limit the registration, as the muscle tissue was not as clear as it would be with other acquisition parameters.

Overall, the main limitation of the proposed method clearly lies in the non-satisfactory capture of individual muscle volume and there are several potential reasons for this to have been found. Firstly, this could have been caused by the propagation of inaccuracies associated with the manual segmentations of the reference images through the registration. However, this aspect is likely to have had a negligible effect since the muscles with high inter-operator variability [15] were discarded at source. More likely, the issue lied in the fact that the muscle-muscle boundaries present a weak grey-level gradient, in contrast to the muscle-fat boundaries, which are shown to have a strong grey-level gradient within the MR images (Figs 1, 2 and 4). Since ShIRT registers grey-level gradients within the inputted images [26], the muscle-fat and muscle-bone boundaries were registered to a higher degree of accuracy than the muscle-muscle boundaries. The use of other MR imaging acquisition settings, such as the Dixon method for fat suppression, could further enhance muscle-muscle boundaries, however, these images were not collected at the time of data acquisition. The use of T1-weighted images has a greater potential as these are by far the most common MR imaging setting seen throughout the clinical domain when assessing soft tissues. This imbalance in the accuracy of the registration of the different tissues is highlighted by the greater RVE of the individual muscles, when compared to the total volume error. Moreover, ShIRT was the only registration algorithm tested. Other available registration algorithms [23–25] could improve the accuracy of the segmentation but will have to be tested on the same dataset. For this reason, the datasets including the input MRI images, the manual segmentations and the ShIRT inputs have been shared here (https://doi.org/10.15131/shef.data.21739733) for future comparison with other registration tools. Another source of error could lie within the optimisation process of the registration parameters (NS and smoothing coefficient) [26]. While in this study these parameters were optimised for the highest overall performance in segmentation accuracy across all considered lower limb muscles, the values could be optimised for the different areas of the limb. This was not implemented in this study as a rewriting of the registration toolkit would be required. The multi-atlas approach was employed to overcome the potential limitations of the registration procedure, incorporating a probabilistic evaluation of which regions of the images belong to each muscle (code available with examples at https://doi.org/10.15131/shef.data.21763982). This method has been used in the assessment of other tissues in the body with good results [30–32]. The method did not have the same impact in this case, most likely due to the sheer number of different muscles assessed, which resulted in a great number of disputed voxels within each target image; a problem which would not be associated with medical image segmentation problems with fewer classes required to be segmented. Though the vast number of disputed voxels was unforeseen, it is logical that there was a large amount of disagreement between automatic segmentations. It has been noted within the literature that this voting system is best suited for a thin layer of disputed voxels surrounding the tissue of interest [31], which was not the case in the automatic segmentations outputted from the inter-subject analyses.

Despite the above limitations, the image registration protocol here proposed proved clearly useful when adopted to generate an augmented imaging database of 69 subjects having a much broader range of muscle volumes and geometries than the original 11 subject database. This result came after removing 41 anatomically unrealistic datasets, which required some manual checking the augmented datasets, suggesting that similar care should be taken if replicating the use of the method. These datasets, made publicly available (augmented images: https://doi.org/10.15131/shef.data.20440164, augmented images segmentations: https://doi.org/10.15131/shef.data.20440203), can be used to train deep learning methods [34, 35]. Machine learning and

deep learning methods are now dominant tools used within the field of medical image segmentation [20, 22, 33]. Where the average DSC found amongst the 23 muscles considered within the present study were found to be around 0.75, considering only the optimal reference subject for each target subject, deep learning methods have been used to segment the lower limb muscles with average DSC between 0.85 [20] and 0.90 [22]. These tools are typically only suitable for studies with extremely large cohorts, but this problem has been alleviated within some medical image analysis fields, such brain tumour assessment [36] and bone segmentation [37], through data augmentation. However, this technique is yet to have been explored for muscle segmentation and the database here presented will hopefully foster efforts in this direction. To the best of our knowledge, in fact, this is the first study providing a vast, multi-operator assessed set of fully segmented, labelled augmented MR imaging sequences of the lower limb. In future work, these augmented datasets will be used to calibrate CNN models, with the potential to increase segmentation accuracy [35, 36] and lead to a solution for the automatic segmentation and characterisation of muscles in vivo.

## Conclusion

This study presented a novel, fully automatic muscle segmentation method using image registration, aimed at segmenting all lower limb muscles simultaneously. The 3D deformable image registration algorithm used in this work is limited in its capacity to perform individual automatic muscle segmentation with a high accuracy. Nevertheless, this approach can be useful to provide total muscle volume and can be used as a tool to increase the number of reference datasets, enabling other methodologies (e.g. learning-based methods) to be explored and properly trained. Explicitly, the publicly available augmented database built in this work would enhance any future study that would aim to use deep learning approaches for the segmentation of muscles from T1-weighted MR images.

## Supporting information

**S1 File. Muscle volumes of the 5 subjects automatically segmented in the study.** Reference muscle volumes for the 23 muscles segmented in this study. Mean and standard deviation are reported.
(PDF)

**S2 File. Sensitivity analysis of the two registration parameters: Nodal spacing, and the smoothing coefficient.** Registration protocol and results for the left to right (intra-subject) analysis, providing the optimal values for the nodal spacing and smoothing coefficient, two registration parameters.
(PDF)

**S3 File. Visualisation of augmented datasets for one target subject.** Display of augmented datasets for one target subject. The image on the left shows a cross- section of the target subject (Subject1) with the manual segmentations for that image shown in green. The 10 images on the right are cross-sections of the augmented datasets, generated when keeping subject 1 as the target for the registration, whilst using the other 10 subjects as the reference dataset. Segmentations are reported in blue. One augmented dataset marked with a red square did not pass the inclusion criteria, due to the discontinuity in the boundary of the body.
(PDF)

**S4 File. Comparisons of muscle volumes within the original and the augmented databases.** Comparisons of muscle volumes within the original (11 subjects) and the augmented (69 virtual subjects) databases. The mean volume ± standard deviation and volume ranges are

reported.
(PDF)

**S5 File. Statistical interpretation of the effect of the pre-processing step used to remove the layer of fat tissue surrounding the muscle tissue.** Statistical analysis and qualitative interpretation of the effects of the pre-processing stage included in the segmentation algorithm.
(PDF)

## Acknowledgments

The authors would like to extend specific thanks to Dr. Erica Montefiori and Dr. Claude Fiifi Hayford for their assistance in this project. Lastly, we are particularly grateful for the participants that volunteered for the study.

## Author Contributions

**Conceptualization:** William H. Henson, Claudia Mazzá, Enrico Dall'Ara.

**Data curation:** William H. Henson, Enrico Dall'Ara.

**Formal analysis:** William H. Henson.

**Funding acquisition:** Claudia Mazzá, Enrico Dall'Ara.

**Investigation:** William H. Henson.

**Methodology:** William H. Henson, Claudia Mazzá, Enrico Dall'Ara.

**Project administration:** Claudia Mazzá.

**Software:** William H. Henson.

**Supervision:** Claudia Mazzá, Enrico Dall'Ara.

**Writing – original draft:** William H. Henson.

**Writing – review & editing:** Claudia Mazzá, Enrico Dall'Ara.

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
