## [Decision Letter · Decision Letter 0]

11 Nov 2022

PONE-D-22-22046Deformable image registration for automatic muscle segmentation and the generation of augmented imaging datasetsPLOS ONE

Dear Dr. Henson,

Thank you for submitting your manuscript to PLOS ONE. After careful consideration, we feel that it has merit but does not fully meet PLOS ONE’s publication criteria as it currently stands. Therefore, we invite you to submit a revised version of the manuscript that addresses the points raised during the review process.

We look forward to receiving your revised manuscript.

Kind regards,

Gernot Reishofer, Ph.D.

Academic Editor

PLOS ONE

Journal Requirements:

The study was partially funded by  Engineering and Physical Sciences Research Council (EPSRC) Frontier Multisim Grant. EP/K03877X/1, CM

EP/S032940/1, CM, ED

https://www.ukri.org/councils/epsrc/

The study was partially funded by Engineering and Physical Sciences Research Council (EPSRC) Frontier Multisim Grant (EP/K03877X/1 and EP/S032940/1). 

However, funding information should not appear in the Acknowledgments section or other areas of your manuscript. We will only publish funding information present in the Funding Statement section of the online submission form. 

The study was partially funded by  Engineering and Physical Sciences Research Council (EPSRC) Frontier Multisim Grant. EP/K03877X/1, CM

EP/S032940/1, CM, ED

https://www.ukri.org/councils/epsrc/

Reviewers' comments:

Reviewer's Responses to Questions

**Comments to the Author**

1. Is the manuscript technically sound, and do the data support the conclusions?

Reviewer #1: Yes

Reviewer #2: Partly

Reviewer #3: Yes

2. Has the statistical analysis been performed appropriately and rigorously? 

Reviewer #1: Yes

Reviewer #2: Yes

Reviewer #3: Yes

3. Have the authors made all data underlying the findings in their manuscript fully available?

Reviewer #1: Yes

Reviewer #2: Yes

Reviewer #3: Yes

4. Is the manuscript presented in an intelligible fashion and written in standard English?

Reviewer #1: Yes

Reviewer #2: Yes

Reviewer #3: Yes

5. Review Comments to the Author

Reviewer #1: The authors present a deformable image registration for automatic muscle segmentation and generation of augmented imaging datasets of the lower extremities. I think, this is generally an important issue to develop automated registration for the quantification of muscle and fat in the thigh and lower leg, especially in the context of sarcopenia and muscle fatty degeneration and associated diseases.

Nevertheless, I have a criticisms or questions about the work.

1.) The introduction is too long, can you please shorten it.

2.) Methods: Why did you use T1 images for the segmentation. Why were MR data not acquired with fat-water separation technique ( e.g., Dixon), since intermuscular fat can be better segmented here?

3.) Methods: Why do you perform this preprocessing step, including the homogenisation of the fatty tissue. Doesn't this make it much harder to detect fatty muscle infiltration? Does not lose the accuracy of the method ?

4.) Results: I don`t feel that the image registration works very well, it seems tob e anatomically not very exact.. are are you sure it can be optimized to the point where it is anatomically accurate enough ?

5.) Results: Do you have an explanation why one of the subjects was the worst performing reference concerning the DSC. Has the subject a high BMI?

6.) Do you have an explanation why the adductor brevis and the recuts femoris made up the outliers concerning RVE..

7.) Is it possible to evalulate more data, to find out in which cases your approach works better?

8.) You state, that this technique can provide muscle volume, but on the other hand it is not possible to provide information about muscle characteristics like is the fatty infiltration whithin the muscle, but this is an important point, as recent research showed thigh intermuscular adipose tissue appears to be a potent muscle variable related to the ability of older adults to move, more than the lean mass. So how can this method really be useful, especially because it is also not quite exact.

Reviewer #2: The authors present a non-rigid registration-based approach to automatic muscle segmentation of 3D MR images of the lower limbs. The resulting deformation fields are used to create new unseen augmented images that could be suitable for training deep learning models in the future. All the data appears to be available online and available for research use (though I encourage the authors to make this info clearer in their manuscript). The work is important in the area of muscle segmentation because it is very time consuming and challenging task to complete manually. The paper is suitably written, and the results are promising.

In summary, my major concern is that an overall DSC of 0.72 is not going to be sufficient for deep learning models to utilise the augmented images in order to improve muscle segmentation literature because it will form the upper limit on the accuracy of the segmentations. According to the authors literature review, other CNN models achieved 0.9 DSC, so 0.72 seems a bit low (though 0.9 DSC seems a bit high, but in accordance with the literature according to the authors own discussion point) with high variability (see whiskers of DSC box plots in Figure 5). If the authors would like to argue that this result is sufficient, they would need to show that the inter-rater variability of their manuals are of a similar level of DSC, but this will require many hours of manual segmentation, thus I suggest doing multi-atlas, improving pre-processing and comparing against a well-known registration framework in addition to the aging SHIRT framework.

Specifically:

1. The accuracy of the results and the methodology of the automatic segmentation seem to hinge around the use of a single atlas approach rather than a multi-atlas approach, where multiple registrations are completed per image (already done) and final segmentations are obtained through some form of voting (e.g. STAPLE etc.) among the registered segmentations using mutual information etc. (see for example reference in the area [1]) This could be why the overall DSCs are low being around 0.72. Could the authors clarify if they used multi-atlas approach for segmentations? If not, I would strongly suggest that they do because it will not only improve the results, but in this era of deep learning, having a multi-atlas approach is a minimum in order to compete and remain relevant.

2. The authors mention in the intro that “An in-house image registration algorithm (Sheffield Image Registration Toolkit, ShIRT) has been used to segment both hard [43] and vascular [42,43] tissues with a high level of accuracy but has not yet been tested in the application of muscle segmentation.”. The authors should:

a. Move this to the methods section as this is among the most critical parts of the paper. The accuracy and quality of the segmentations depend on this info and it’s methodology explanation is fractured in the paper.

b. This is very little mention of the mechanism behind the registration algorithm and relevant citations of the registration type utilised unless I missed it. Is it based around optical flow or free form deformation etc?

c. Justify why other registration algorithms were not explored or compared against. Very mature and open source (which SHIRT does not appear to be?) and most importantly parallel frameworks such as VoxelMorph, Elastix, NiftiReg or ITK.

d. Given the low DSC of the results, and the age of SHIRT, I’m inclined to request that the authors add a comparison to one of these registration frameworks, unless the multi-atlas approach is able to provide a reasonable improvement of the results.

3. The authors mention “To the best of the authors’ knowledge, this study represents the first attempt to segment complete 3D muscle geometry of many individual muscles simultaneously using deformable image registration while using different subjects as the reference.” Only in the discussion section. I urge the authors to add more clearer novelty statement such as this earlier in the manuscript, such as the end of the intro where the aim is first introduced.

4. It didn’t seem that the authors utilised the remaining 6 of the 11 patients for segmentation (5 were manually segmented and used for evaluation). Could not the 5 manual segmentations used to bootstrap further manual (or automatic) segmentations of the remaining 6 subjects or were these only used for generating the augmentation?

5. I could not see the use of bias field correction methods such as N4. It would seem to be that MR images of such wide field of view would be significantly affected by bias fields. Can the authors comment if they used it? If not I would strongly suggest using it because it could also improve the segmentation accuracy quite a bit since registrations will be more accurate.

6. Table 1 should really be in the results section because it presents the results of the manual segmentations.

7. The literature review seems rushed with many references being cited for single points. For example: “The large variability of muscle volume and geometry within the lower limb skeletal muscles between subjects, even within cohorts with similar anthropometric characteristics, limits the application of SSM to segment these muscles [15,29,30,31].” And “Many different automatic segmentation methods have been investigated within the literature in recent years to replace the manual approach [21,24,25,27].” It would be good to expand a few of these types of sentences to give more details about some of these works.

8. I also could not determine if there was an initialisation of the deformable registration. Was it an affine or rigid registration? What was the optimizer and similarity metric used? These technical details of the registration methods are important for reproducibility and I would request a sub-section in the methods or results dedicated to it.

Apologies in advanced if I missed anything.

References

[1] S. Klein, U. A. van der Heide, I. M. Lips, M. van Vulpen, M. Staring, and J. P. W. Pluim, “Automatic segmentation of the prostate in 3D MR images by atlas matching using localized mutual information,” Medical Physics, vol. 35, no. 4, pp. 1407–1417, 2008, doi: 10.1118/1.2842076.

Reviewer #3: Muscle segmentation is relevant to many medical fields (aging, musculoskeletal and neuromusculoskeletal disorders). The limitations of traditional segmentation are pointed out clearly (excessive interaction time, low inter-operator agreement). Recent advances in deep learning-based methods have brought about questions regarding the underlying datasets. The gerneration of augmented datasets is therefore a very important activity to further increase the power of such methods. Both visual and statistical inspection of the results show moderate results with room for improvement in certain areas. As indicated in the conclusion of the paper, comparing ShIRT to deep learning-based methods (e.g., VoxelMorph) may provide valuable insights.

6. PLOS authors have the option to publish the peer review history of their article (what does this mean?). If published, this will include your full peer review and any attached files.

Reviewer #1: No

Reviewer #2: No

Reviewer #3: **Yes: **Bernhard Schenkenfelder

---

## [Author Response · Author response to Decision Letter 0]

21 Dec 2022

C1. The authors present a deformable image registration for automatic muscle segmentation and generation of augmented imaging datasets of the lower extremities. I think, this is generally an important issue to develop automated registration for the quantification of muscle and fat in the thigh and lower leg, especially in the context of sarcopenia and muscle fatty degeneration and associated diseases. Nevertheless, I have a criticisms or questions about the work.

A1. We thank the reviewer for recognising the importance of this problem and agree with the research areas they highlight that could be aided with an approach such as this.

C1.1. The introduction is too long, can you please shorten it.

A1.1. The introduction has been shortened in the aspects that are not directly related. With this reduction, we have expanded the sections that are more relevant, focussing more on registration and deep learning methodologies. In particular, we shortened the paragraph beginning on line 118.

C1.2. Methods: Why did you use T1 images for the segmentation. Why were MR data not acquired with fat-water separation technique ( e.g., Dixon), since intermuscular fat can be better segmented here?

A1.2. While the fat suppression techniques noted would have simplified the registration, we tested the algorithm based on the more broadly used T1-weighted images. Results would be expected to improve with fat suppressed images and this has been recognised in the limitations section of the discussion (lines 717-721 & 754-758).

C1.3. Methods: Why do you perform this pre-processing step, including the homogenisation of the fatty tissue. Doesn't this make it much harder to detect fatty muscle infiltration? Does not lose the accuracy of the method?

A1.3. In a preliminary study, we registered the original images and found that the fat surrounding the muscle tissue skewed the registration. We found that in most of the tested subject combinations, the segmentation accuracy was increased after pre-processing the images in this way, and no combination presented a reduction in segmentation accuracy after the pre-processing step. The paper has been adjusted to highlight this (lines 310-313), including some additional supplementary material (number 5).

C1.4. Results: I don`t feel that the image registration works very well, it seems to be anatomically not very exact.. are you sure it can be optimized to the point where it is anatomically accurate enough?

A1.4. We agree that the registration algorithm is not working as accurately as we would have hoped for every subject. We highlight this in the results section, where the individual muscle segmentation is stated not to have been performed to a satisfactory standard, but the total muscle volume can be gathered with a high level of accuracy. 

We have added a multi-atlas approach to explore potential improvements and reported the results in the manuscript (lines 370-373, 395-399, 407-411). While by using this approach the results improved, they haven’t reached the accuracies obtained by registering left vs right limbs. 

C1.5. Results: Do you have an explanation why one of the subjects was the worst performing reference concerning the DSC. Has the subject a high BMI?

A1.5. The BMI of the investigated subjects are very similar, but we identified that there was a large difference in age (16 years). This could explain the difference in muscle quality [1], but due to the low sample size we could not identify if this was the issue. The paper has been amended to reflect this point (see lines 701-707).

[1] R. A. Fielding, B. Vellas, W.J. Evans, et al., Sarcopenia: an undiagnosed condition in older adults. Current consensus definition: prevalence, etiology, and consequences. International working group on sarcopenia. JAMDA, 2011, vol 12, issue 4, pages 249-256. DOI: https://doi.org/10.1016/j.jamda.2011.01.003

C1.6. Do you have an explanation why the adductor brevis and the recuts femoris made up the outliers concerning RVE..

A1.6. Apologies, this was labelled incorrectly (it should not be the adductor brevis, should be the tensor fascia latae). These were muscles of high variability within the cohort, meaning that the registration struggled to overcome the variability. 

Thank you for raising this point. The paper has been updated (see line 514-521), with table 1 changed to reflect the variability of muscle volumes for the 23 muscles included.

C1.7. Is it possible to evaluate more data, to find out in which cases your approach works better?

A1.7. We registered the left and right limbs of the considered subjects as we expect that the variability in muscle structure between contralateral limbs to be the minimum possible. We believe, therefore, that these results would be the upper threshold for segmentation accuracy when using an image registration algorithm such as the one here adopted. We suggest (lines 583-587) that the next step would be to move toward other approaches and use elastic registration to aid in those applications.

C1.8. You state, that this technique can provide muscle volume, but on the other hand it is not possible to provide information about muscle characteristics like is the fatty infiltration within the muscle, but this is an important point, as recent research showed thigh intermuscular adipose tissue appears to be a potent muscle variable related to the ability of older adults to move, more than the lean mass. So how can this method really be useful, especially because it is also not quite exact.

A1.8. We agree with your point. The approach could be useful for: 

1) accurate segmentations of the total muscle body (average volume error of 8%). Although the algorithm has not been tested for evaluating intermuscular fat infiltration, this could be isolated from the segmentation of the total muscle body.

2) for individual muscle segmentation using the contralateral limb as a reference, halving the time required to gather individual muscle segmentation with reasonable accuracy. 

3) in the generation of augmented datasets to facilitate the exploration of other, learning-based methods.

------------

C2. In summary, my major concern is that an overall DSC of 0.72 is not going to be sufficient for deep learning models to utilise the augmented images in order to improve muscle segmentation literature because it will form the upper limit on the accuracy of the segmentations. According to the authors literature review, other CNN models achieved 0.9 DSC, so 0.72 seems a bit low (though 0.9 DSC seems a bit high, but in accordance with the literature according to the authors own discussion point) with high variability (see whiskers of DSC box plots in Figure 5). If the authors would like to argue that this result is sufficient, they would need to show that the inter-rater variability of their manuals are of a similar level of DSC, but this will require many hours of manual segmentation, thus I suggest doing multi-atlas, improving pre-processing and comparing against a well-known registration framework in addition to the aging SHIRT framework.

A2. We feel that there might be some confusion around this point. The augmented images (available online through figshare) are not segmented with a DSC of 0.72. Indeed, the augmented images and their segmentations are gathered by applying the same deformation to both the reference subject’s images and segmentation, meaning that the segmentation accuracy of the augmented images is equivalent to the manual process itself (i.e. the current gold standard). In fact, if the registration would provide the perfect match, the added virtual cohort would be the same as the starting dataset, bringing no variability. 

C2.1. The accuracy of the results and the methodology of the automatic segmentation seem to hinge around the use of a single atlas approach rather than a multi-atlas approach, where multiple registrations are completed per image (already done) and final segmentations are obtained through some form of voting (e.g. STAPLE etc.) among the registered segmentations using mutual information etc. (see for example reference in the area [1]) This could be why the overall DSCs are low being around 0.72. Could the authors clarify if they used multi-atlas approach for segmentations? If not, I would strongly suggest that they do because it will not only improve the results, but in this era of deep learning, having a multi-atlas approach is a minimum in order to compete and remain relevant.

A2.1. Our original approach was based on single atlas registrations. We have implemented a multi-atlas post-processing step similar to the one suggested (STAPLE). This approach increased the segmentation accuracy slightly. There was an increase in the segmentation accuracy, but only moderate due to a low agreement between the segmentations, probably resulting from the different single atlas registrations. This finding furthers the point that other methods should be adopted for automatic muscle segmentation, such as deep learning rather than registration-based approaches.

The results have been added and discussed in the manuscript.

a. Move this to the methods section as this is among the most critical parts of the paper. The accuracy and quality of the segmentations depend on this info and it’s methodology explanation is fractured in the paper.

b. This is very little mention of the mechanism behind the registration algorithm and relevant citations of the registration type utilised unless I missed it. Is it based around optical flow or free form deformation etc?

c. Justify why other registration algorithms were not explored or compared against. Very mature and open source (which SHIRT does not appear to be?) and most importantly parallel frameworks such as VoxelMorph, Elastix, NiftiReg or ITK.

d. Given the low DSC of the results, and the age of SHIRT, I’m inclined to request that the authors add a comparison to one of these registration frameworks, unless the multi-atlas approach is able to provide a reasonable improvement of the results.

A2.2. We thank the reviewer for the detailed and very useful suggestions. We have acted upon suggestions a and b (lines 310-333), c (lines 760-764), and d (with the inclusion of the multi-atlas approach & amendments to the discussion: lines 772-798).

C2.3. The authors mention “To the best of the authors’ knowledge, this study represents the first attempt to segment complete 3D muscle geometry of many individual muscles simultaneously using deformable image registration while using different subjects as the reference.” Only in the discussion section. I urge the authors to add more clearer novelty statement such as this earlier in the manuscript, such as the end of the intro where the aim is first introduced.

A2.3. We have acted upon this suggestion (throughout introduction but specifically lines 208-213). Thank you for the suggestion.

C2.4. It didn’t seem that the authors utilised the remaining 6 of the 11 patients for segmentation (5 were manually segmented and used for evaluation). Could not the 5 manual segmentations used to bootstrap further manual (or automatic) segmentations of the remaining 6 subjects or were these only used for generating the augmentation?

A2.4. The other 6/11 subjects were used to generate the augmented datasets. To this end, they have been fully manually segmented. These other subjects could have been used in the multi-atlas approach, but this would further increase the already large number of disputed voxels and this is addressed in the limitations of the work (lines 772-798).

C2.5. I could not see the use of bias field correction methods such as N4. It would seem to be that MR images of such wide field of view would be significantly affected by bias fields. Can the authors comment if they used it? If not I would strongly suggest using it because it could also improve the segmentation accuracy quite a bit since registrations will be more accurate.

A2.5. The paper has been adjusted in response to this comment (lines 273-275). The MR image sequence combination script removed those images that were the most affected by MR imaging bias (upper and lower 5-10 images in each sequence). A bias field correction algorithm was tested but did not alter the images significantly.

C2.6. Table 1 should really be in the results section because it presents the results of the manual segmentations.

A2.6. This was a result from a previous study and the authors feel it would be misleading to include this table in the results section of the study. The table is required in the paper as it justifies some of the discussion points and identifies all the muscles assessed within the paper.

C2.7. The literature review seems rushed with many references being cited for single points. For example: “The large variability of muscle volume and geometry within the lower limb skeletal muscles between subjects, even within cohorts with similar anthropometric characteristics, limits the application of SSM to segment these muscles [15,29,30,31].” And “Many different automatic segmentation methods have been investigated within the literature in recent years to replace the manual approach [21,24,25,27].” It would be good to expand a few of these types of sentences to give more details about some of these works.

A2.7. The literature review has been altered in response to this suggestion (lines 76-106), reducing unnecessary sections and expanding on the relevant topics (image registration, multi-atlas, and deep learning approaches). Thank you for pointing this out.

C2.8. I also could not determine if there was an initialisation of the deformable registration. Was it an affine or rigid registration? What was the optimizer and similarity metric used? These technical details of the registration methods are important for reproducibility and I would request a sub-section in the methods or results dedicated to it.

A2.8. This has been clarified (lines 364-366). Thank you for the suggestion.

 

------------

C3. Muscle segmentation is relevant to many medical fields (aging, musculoskeletal and neuromusculoskeletal disorders). The limitations of traditional segmentation are pointed out clearly (excessive interaction time, low inter-operator agreement). Recent advances in deep learning-based methods have brought about questions regarding the underlying datasets. The gerneration of augmented datasets is therefore a very important activity to further increase the power of such methods. Both visual and statistical inspection of the results show moderate results with room for improvement in certain areas. As indicated in the conclusion of the paper, comparing ShIRT to deep learning-based methods (e.g., VoxelMorph) may provide valuable insights.

A3. We appreciate the kind words, and we are looking toward the suggestions made in the future, and results area already promising.

---

## [Decision Letter · Decision Letter 1]

15 Feb 2023

Deformable image registration based on single or multi-atlas methods for automatic muscle segmentation and the generation of augmented imaging datasets

PONE-D-22-22046R1

Dear Dr. Henson,

We’re pleased to inform you that your manuscript has been judged scientifically suitable for publication and will be formally accepted for publication once it meets all outstanding technical requirements.

Kind regards,

Gernot Reishofer, Ph.D.

Academic Editor

PLOS ONE

Additional Editor Comments (optional):

Reviewers' comments:

Reviewer's Responses to Questions

**Comments to the Author**

1. If the authors have adequately addressed your comments raised in a previous round of review and you feel that this manuscript is now acceptable for publication, you may indicate that here to bypass the “Comments to the Author” section, enter your conflict of interest statement in the “Confidential to Editor” section, and submit your "Accept" recommendation.

Reviewer #1: All comments have been addressed

Reviewer #3: All comments have been addressed

2. Is the manuscript technically sound, and do the data support the conclusions?

Reviewer #1: Yes

Reviewer #3: Yes

3. Has the statistical analysis been performed appropriately and rigorously? 

Reviewer #1: Yes

Reviewer #3: Yes

4. Have the authors made all data underlying the findings in their manuscript fully available?

Reviewer #1: Yes

Reviewer #3: Yes

5. Is the manuscript presented in an intelligible fashion and written in standard English?

Reviewer #1: Yes

Reviewer #3: Yes

6. Review Comments to the Author

Reviewer #1: Although the segmentation seems to work not really exact as the authors stated by themselves, it is an interesting paper and an innovative approach.

All comments have been adressed.

Reviewer #3: When generating augmented imaging datasets, are you planning on collecting/providing metadata? Something along the lines of https://dl.acm.org/doi/10.1145/3458723

I'm looking forward to hearing about your future work.

7. PLOS authors have the option to publish the peer review history of their article (what does this mean?). If published, this will include your full peer review and any attached files.

Reviewer #1: No

Reviewer #3: No

---

## [Editor Report · Acceptance letter]

28 Feb 2023

PONE-D-22-22046R1 

Deformable image registration based on single or multi-atlas methods for automatic muscle segmentation and the generation of augmented imaging datasets 

Dear Dr. Henson:

I'm pleased to inform you that your manuscript has been deemed suitable for publication in PLOS ONE. Congratulations! Your manuscript is now with our production department. 

Kind regards, 

on behalf of

Dr. Gernot Reishofer 

Academic Editor

PLOS ONE